

# A low-cost benthic incubation chamber for *in-situ* community metabolism measurements

Jennifer Mallon[1], Anastazia T. Banaszak[2], Lauren Donachie[3], Dan Exton[3], Tyler Cyronak[4], Thorsten Balke[1] and Adrian M. Bass[1]

[1] School of Geographical and Earth Sciences, University of Glasgow, Glasgow, United Kingdom
[2] Unidad Académica de Sistemas Arrecifales, Universidad Nacional Autónoma de México, Puerto Morelos, Quintana Roo, Mexico
[3] Operation Wallacea, Spilsby, Lincolnshire, United Kingdom
[4] Department of Marine and Environmental Sciences, Halmos College of Natural Sciences and Oceanography, Nova Southeastern University, Dania Beach, Florida, United States

Corresponding author
Jennifer Mallon,
j.mallon.1@research.gla.ac.uk,
jmallon967@gmail.com

## ABSTRACT

Benthic incubation chambers facilitate *in-situ* metabolism studies in shallow water environments. They are used to isolate the water surrounding a study organism or community so that changes in water chemistry can be quantified to characterise physiological processes such as photosynthesis, respiration, and calcification. Such field measurements capture the biological processes taking place within the benthic community while incorporating the influence of environmental variables that are often difficult to recreate in *ex-situ* settings. Variations in benthic chamber designs have evolved for a range of applications. In this study, we built upon previous designs to create a novel chamber, which is (1) low-cost and assembled without specialised equipment, (2) easily reproducible, (3) minimally invasive, (4) adaptable to varied substrates, and (5) comparable with other available designs in performance. We tested the design in the laboratory and field and found that it achieved the outlined objectives. Using non-specialised materials, we were able to construct the chamber at a low cost (under $20 USD per unit), while maintaining similar performance and reproducibility with that of existing designs. Laboratory and field tests demonstrated minimal leakage (2.08 ± 0.78% water exchange over 4 h) and acceptable light transmission (86.9 ± 1.9%), results comparable to those reported for other chambers. In the field, chambers were deployed in a shallow coastal environment in Akumal, Mexico, to measure productivity of seagrass, and coral-, algae-, and sand-dominated reef patches. In both case studies, production rates aligned with those of comparable benthic chamber deployments in the literature and followed established trends with light, the primary driver of benthic metabolism, indicating robust performance under field conditions. We demonstrate that our low-cost benthic chamber design uses locally accessible and minimal resources, is adaptable for a variety of field settings, and can be used to collect reliable and repeatable benthic metabolism data. This chamber has the potential to broaden accessibility and applications of *in-situ* incubations for future studies.

## INTRODUCTION

Quantifying the relative balance of autotrophic and heterotrophic processes taking place within a benthic community provides insight into ecosystem functioning and species composition (*Albright et al., 2015*; *Cyronak et al., 2018*). As coastal ecosystems undergo degradation, understanding ecosystem capacity for oxygen production and carbon cycling through photosynthesis is critical. For example, defining ecosystem productivity can support conservation of high 'blue carbon value' ecosystems (*Duarte et al., 2010*; *Duarte, Sintes & Marbà, 2013*) and characterisation of community metabolism facilitates geographical and temporal comparisons of ecological function, which otherwise might be logistically limited (*Cyronak et al., 2018*; *Lange, Perry & Alvarez-Filip, 2020*). Measurements of *in-situ* community metabolism capture individual rates of productivity for functional groups and species, as well as the interactive physiological processes taking place within the entire benthic community, rather than the simplified, reconstructed communities often used in *ex-situ* measurements. Such measures offer important insight into the critical role of coastal marine ecosystems for carbon capture and cycling.

Rates of photosynthesis and respiration can be measured directly from fluxes in dissolved oxygen concentrations to calculate the net community productivity (NCP) taking place within a benthic community. The development of *in-situ* gear to incubate benthic organisms and communities has been guided by distinct research applications. Ecosystem metabolism can be measured *via* three main approaches in the field; benthic boundary layer and eddy covariance (*e.g.*, *Long, Berg & Falter, 2015*; *Takeshita et al., 2016*; *Berg et al., 2019*, *2022*), flow respirometry using Lagrangian and Eularian adaptations (*e.g.*, *Barnes, 1983*; *Falter et al., 2008*; *Shaw et al., 2014*), and enclosed incubations (see examples in Table 1). This study focusses on the incubation method as a straightforward approach for deriving metabolic rates of single organisms or benthic communities in the field.

Benthic chambers can be deployed in the field to contain sediments, corals, seagrasses, and other biota, either as communities or as individual organisms, for periods of hours to days to capture *in-situ* measurements of community or organismal metabolic rates (*e.g. Huettel & Gust, 1992*; *Yates & Halley, 2003*; *Wild et al., 2005*; *Murphy et al., 2012*; *Camp et al., 2015*; *van Heuven et al., 2018*; *Roth et al., 2019*). Standard benthic chamber designs consist of a tent or dome made from a rigid, transparent container with a circulation pump (to mimic natural water flow), and a sampling port (*e.g.*, *Roth et al., 2019*, Table 1). Some chambers can only be deployed in areas where substrate is suitable for the chamber base to be inserted, for example, often the base must be buried in sand or sediments to create an effective seal (*e.g.*, *Olivé et al., 2016*). However, the technology has evolved to encompass a range of applications from smaller, single organism chambers to larger community enclosures (Table 1). For example, high precision, real-time, *in-situ* metabolism measurements of small surface areas of coral have been made possible by the development of a high-tech adaptation of the benthic chamber concept: the Coral *in situ*
**Table 1 Summary of the key features of selected benthic chamber designs since 2000.** Benthic chamber designs were selected to incorporate a size gradient ranging from sub-organism to community incubations. We selected studies which evaluate the performance of the chambers. This table is not exhaustive and only provides a summary of chamber types. The chamber of *Yates & Halley, (2003)* is reprinted by permission from Springer Nature Coral Reefs, Measuring coral reef community metabolism using new benthic chamber technology, *Yates & Halley, (2003)*.

| | Name | Scale | Cost (USD) | Leakage | Light trans. | Strengths | Limitations |
|---|---|---|---|---|---|---|---|
|  | Coral *In-Situ* Metabolism and Energetics (CISME), (*Romanó de Orte et al., 2021*; *Murphy et al., 2012*). | <1 individual | $32,000 | Unreported, expected minimal | 0% (Artificial light) | High accuracy and precision. Instantaneous measurements. | Expensive and specialised. Very small areas can be incubated (24.5 cm$^2$). |
|  | Flexi-Chamber (*Camp et al., 2015*). | Single | <$20 | None-to-minimal | 84% | Low-cost and easily sourced materials. Flexible material maintains water movement. | Cable tie closure around the base is not suitable for all benthic organisms. |
|  | Flexi-community benthic chamber for remote locations, (*this study*). | Single small community | <$20 | 2% over 4 h | 89% | Low-cost, easily sourced materials. Adaptable design. Flexible material maintains water movement. | Under some conditions seal is not as reliable as other methods. |
|  | *In-situ* chambers for measuring biogeochemical fluxes (*Roth et al., 2019*). | Single small community | ~$250 | 5.3% within 6 h (reef sands) 12.4% within 6 h (rocky) | 92% | Can be deployed on hard substrate or sediments. | Requires circulation pump. Bespoke design requires specific materials. |
|  | Submersible Habitat for Analyzing Reef Quality (SHARQ) (*Yates & Halley, 2003*). | Larger communities (~12 m$^2$ planar surface area) | Not reported | Variable depending on substrate type | >71% | Can incubate entire communities. | Requires circulation pump. Bespoke design requires specific materials. |

Metabolism and Energetics (CISME) system, described in *Murphy et al. (2012)*. CISME is a specialised underwater respirometer that incubates small areas (~24.5 cm$^2$) of live coral to measure pH, dissolved oxygen, and temperature changes. The electronic housing has waterproof cables supplying power, a recirculation pump, and LED control, while a water sample loop holds incubation samples so that total alkalinity can be used to derive calcification rates.

At the opposite end of the size-spectrum, large tent-like structures have been developed to incubate entire patches of benthic communities (surface areas of several m$^2$). For example, the Submersible Habitat for Analysing Reef Quality (SHARQ) developed by
*Yates & Halley (2003)* and other large tent enclosures such as the one described by *van Heuven et al. (2018)*, allow for measurements of seawater chemistry and can even be adapted for *in-situ* experiments adding $CO_2$ or other treatments, *e.g.*, *Kline et al. (2012)*. The SHARQ and similar chamber designs (*e.g.*, *van Heuven et al., 2018*) incorporate a submersible circulation pump to ensure turbulent flow within the incubated area. Flow is a primary driver of benthic metabolism and regulates organism response to environmental effects such as ocean acidification (*Comeau et al., 2014*, *2019*). The costs of building such chambers are moderate to high, and multiple replicate chambers for parallel incubations further raise the costs, while specialised materials may be difficult to source in some parts of the world where coral reefs and seagrasses are most abundant. Submersible pumps, bespoke parts, and other expensive materials create an economic barrier to the *in-situ* incubation chamber technique. To address these limitations, the Flexi-chamber, developed by *Camp et al. (2015)*, requires minimal specialised materials and can be constructed at a low cost (<USD $20). The Flexi-chamber is made of a flexible plastic bag with a sampling valve installed, which is attached to the base of the study organism with a cable tie. Both the bag and valve were sourced from a medical supply store. The Flexi-chamber can be used to measure metabolism for single organisms, and it is ideal for branching corals, or organisms with morphological formations which are raised from the substate. The Flexi-chamber was robustly tested in the laboratory and field and performed in line with other incubation chamber designs (*Camp et al., 2015*). However, the chamber can only be used on substrates suitable for a cable tie attachment. For flatter substrates, a domed benthic chamber design is more effective. The medium size chamber by *Roth et al. (2019)* can be placed over the substrate and was successfully produced at a lower cost (~$250 USD) than previous designs, while maintaining the efficiency and precision of more expensive equipment. However, the design requires a bespoke rigid acrylic cylinder and a circulation pump to recreate water movement inside the chamber, incurring a higher cost.

To contribute to the array of chambers currently available for field incubations, we designed a chamber prototype, which was both low cost and suitable for deployment on complex substrates incorporating small benthic communities to accurately measure community metabolism. Our objectives were to design a benthic chamber that is; (1) low cost and easily constructed without specialised equipment, (2) reproducible for scientific soundness, (3) minimally invasive to reduce any impact on incubated organisms, (4) adaptable for use in a range of substrate types and underwater environments, and (5) comparable with other chamber designs such as those described in Table 1.

## MATERIALS AND METHODS

### Study location

Experiments were carried out in the School of Geographical and Earth Sciences Laboratory at the University of Glasgow, the Scottish Centre for Ecology and the Natural Environment (SCENE) fieldwork site, and in Akumal Bay Natural Refuge Area, Mexico. SCENE is located on the eastern shore of Loch Lomond in west central Scotland (56°07′41.1″ N, 4°36′48.4″ W). Akumal is a small coastal town on the Caribbean coast of the Yucatan peninsula, with relatively limited access to specialised materials. The Natural Refuge of

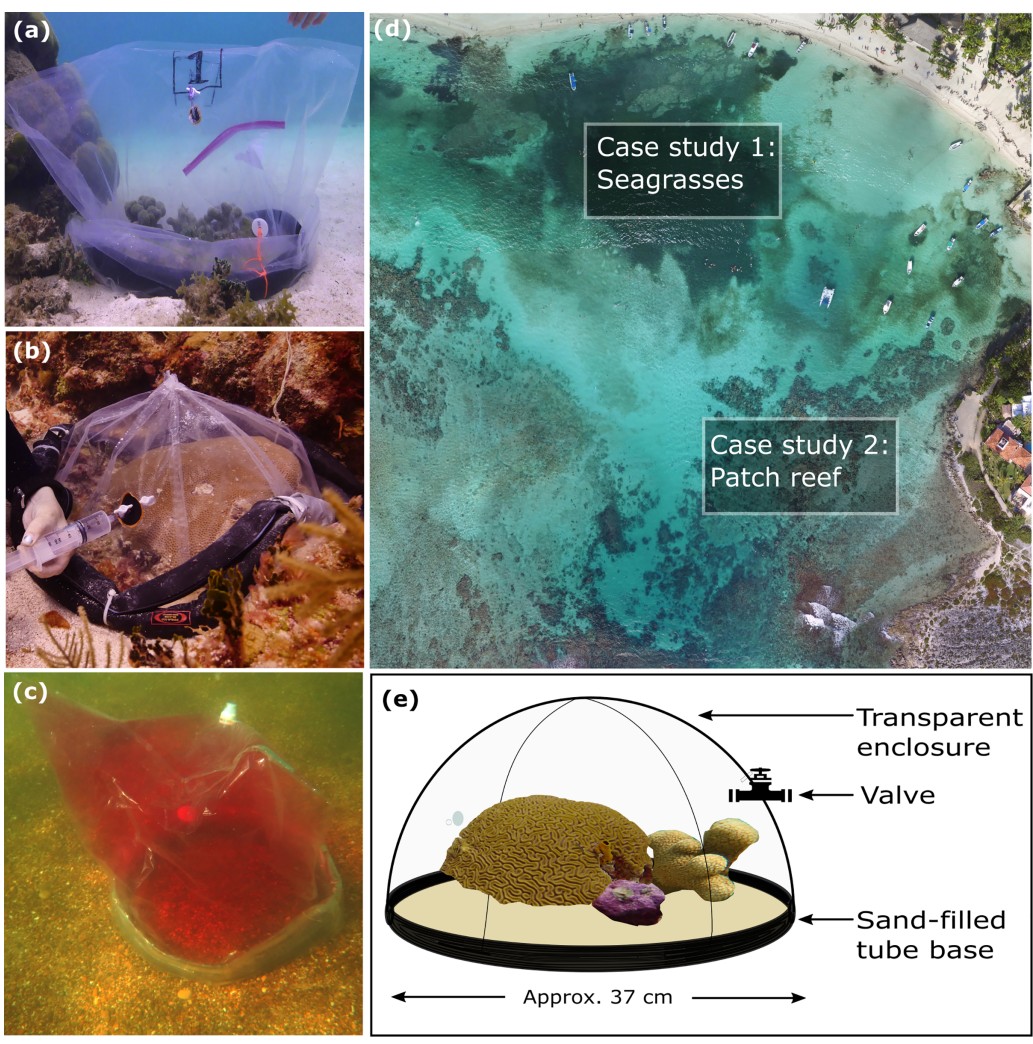

**Figure 1** **Visual summary of field deployed benthic chambers for case studies 1 and 2.** Photographs depicting: (A) tall version of the chamber constructed using the unchanged plastic bag; (B) low-profile chamber created by cutting the plastic bag into panels and heat sealing in a dome form, for use in high-energy environments; (C) visual assessment of leakage and mixing from the chamber using dispersal of red food dye injected into the chamber; (D) aerial image taken above Akumal Bay showing the sites where chambers were deployed for case studies, photo credit: Edgar Escalante Mancera and Miguel Ángel Gómez Reali; and (E) a schematic diagram of the chamber prototype developed in this study.

Akumal Bay (20°23′42.2″ N, 87°18′49.8″ W) is a protected area within the Mexican Caribbean Biosphere Reserve and consists of a semi-enclosed lagoon with patches of seagrass and coral ecosystems (Fig. 1A). Field experiments in Akumal were approved by the Comisión Nacional de Áreas Naturales Protegidas (CONANP F009.DRBCM/240/2019).

## Chamber design, materials, construction, and costs
The benthic chamber design consists of three key components: (1) a water-tight polyethylene tent, (2) a sampling valve, and (3) a heavy yet malleable circular base for

**Table 2 Summary of key chamber design components.**

| | Key features | Materials and costs (USD) | Adaptations |
|---|---|---|---|
| Tent enclosure | Plastic retains upright structure due to neutral buoyancy, while its flexibility allows movement with water, facilitating mixing and flow. If needed, watertight glue or silicone can be used along the vertical seams to provide a more rigid upright structure. | Transparent polyethylene bags were sourced from local food packaging and grocery stores for $0.25 per unit in Mexico, and for approx. $1.80 per unit in the UK. Glue and silicone for re-sealing were sourced at local hardware stores for <$5, with an estimated cost of $0.50 per chamber. | Size of plastic bag can be changed to incubate different volumes of water or surface areas of substrate. Enclosure transparency or colour filtration can be achieved with different plastics. The shape of chamber can be adapted for different settings. |
| Sampling valve | A valve was installed into each chamber for the extraction of water samples with syringes. Installation involved a pinprick hole in the plastic, with a Luer-lock valve inserted and attached to a second Luer-lock valve on the inside of the chamber. | The valves tested in this study were 3-way Luer-lock valves, priced at $1.12 per unit. In lieu of valves, sports bottle caps can be implemented at a similar cost. | A sampling port of any size can be installed, depending on study requirements. 3-way valves can facilitate the addition of experimental treatments. The location of the valve on the chamber could impact mixing. |
| Base | The heavy chamber base is placed on top of the plastic enclosure skirt to create a seal with the substrate. The base is malleable so that it can be moulded to the complexity of solid substrates, and in the case of soft substrata, buried as needed. | Bicycle inner tubes were cut to size and filled with sand and small fishing weights. After filling, the tubes were sealed using bicycle tyre patches and glue. As the used bicycle tubes were donated there was no cost. New inner tubes cost ~$4 each. Fishing weights $3 per kg. | Base weight can be adjusted as needed, depending on the environmental conditions. Stones can be used in lieu of fishing weights. |

maintaining the tent enclosure seal against the substrate (Fig. 1, Table 2). The tent is made from a polyethylene plastic bag. It is possible to keep the shape of the plastic bag intact, by simply cutting in a straight line at the bottom of the bag for the desired height, allowing an extra 10–15 cm at the bottom to allow the circular malleable base to sit on top. Alternatively, a polyethylene bag or sheet can be cut and re-sealed using a heat sealer into any desired enclosure shape, e.g., dome or pyramid. For case study 1 the bottom of the bag was folded to create a 'skirt' for the base to sit on. In case study 2 a dome shaped benthic chamber was used (Figs. 1B and 1E), which was made by cutting the polyethylene bag into panels and re-sealing using a heat sealer. We used a food packaging heat sealer to join the panels. The valve was inserted by creating a small incision in the enclosure tent and inserting one valve on the inside and securing it with the Luer lock connector of a second valve on the inside. A small bicycle tube repair patch was used to create a solid base for the valve to be inserted through. For the base, bicycle inner tubes were filled with sand and fishing weights to create a heavy, malleable base, and re-sealed using bicycle tyre repair kits found at a local hardware store. All materials to build the chamber were sourced locally. Transparent polyethylene for the tent enclosure was bought from a local food packaging company, Luer-lock valves were sourced in the local pharmacy and pre-used bicycle tyre inner tubes were donated by the local community. The materials used and their associated costs are summarised in Table 2.

## Design validation

Three key characteristics were investigated to validate the reproducibility and correct functioning of the chambers: (1) leakage of water into/out of the chamber, (2) light transmission across the plastic membrane, and (3) temperature stability within the chamber.

### 1. Leakage

To test for leaks, non-toxic red food dye was inserted using a syringe (20 ml) into a chamber deployed over sediments in shallow water at each of our field sites. Visual surveys were conducted by two snorkellers who monitored the water outside of the chamber for any obvious visible leaks of red-dyed water for 30 min (Fig. 1C). An underwater camera (Canon Powershot S120) was attached to a dive weight and placed on the substrate outside the chamber for the duration of the 30-min deployments, and the video was reviewed for any red dye leakage around the base of the chamber. This process was repeated 3 times before case studies and before lab testing. Following this initial coarse assessment, a 90-litre aquarium tank was filled with artificial salt water (tap water prepared with Marine salts at a salinity of 21.5 ± 4.7 ppt) and field water movement was simulated using 4 circulation pumps (Reef Tide 4000s 12 v DC Wavemaker Pump, UK) positioned on the sides of the tank to create a zig-zag water motion across the tank. A prototype of the benthic chamber was placed in the tank and a super-saturated saline solution was inserted using a plastic syringe connected to the valve of the benthic chamber so that the water inside the chamber reached a salinity of 38.3 ± 5.7 ppt. Salinity was measured once per minute during the 4-h incubation, inside and outside the benthic chamber using two cross-calibrated salinity loggers (Onset Hobo U24-002-C Saltwater Conductivity/Salinity Data Logger). The leakage test was repeated 3 times, and for each repeat a different chamber was used.

### 2. Light attenuation

A chamber was deployed over sediments in Akumal Bay at a depth of 2 m for 3 days between 07:00 and 19:00 h in August 2019. A different chamber was used for each day of the experiment ($n = 3$). Photosynthetically active radiation (PAR) was measured simultaneously inside and outside the chamber with two submersible sensors (Odyssey Submersible PAR Logger) attached to a 4 kg dive weight and positioned inside and outside of each chamber to log light each minute ($\mu$mol photons m$^{-2}$ s$^{-1}$). The sensors were calibrated against a recently calibrated Licor quantum sensor (LI-190; LI-COR Biosciences, Lincoln, NE, USA). Additionally, light transmission (%) between the photosynthetically relevant wavelengths of 300–800 nm was measured through the plastic membrane (Shimadzu UV-3600 UV-Vis).

### 3. Temperature

Changes to the temperature of water inside the chamber were measured to demonstrate accumulation of heat, which we would expect to see if the water is not mixing well. Temperature inside the chambers was collected using a handheld probe (Pro DSS

multiparameter; YSI, Yellow Springs, OH, US) at the start and end of field deployments of 9 chambers in tropical conditions over 4 days in case study 2. A separate 4-h deployment in cold water conditions, at the SCENE fieldwork site in Loch Lomond was also conducted in August 2021 and temperature was continuously logged for the duration of the incubation (per minute) inside and outside of the chamber using temperature loggers (Onset HOBO UA-002-08 Pendant 8K Light and Temperature) with an accuracy of ± 0.5 °C.

## Field operation of the benthic chamber

Field deployments over benthic communities were conducted to validate the reproducibility and adaptability of the chambers. Two case studies were conducted in Akumal Bay (Fig. 1D), one with incubations of seagrass patches and the other over coral reef communities (total of 41 deployments). Chambers were installed by divers and snorkellers. For all field incubations, water samples were extracted at the start and end of 1.5 to 3.5-h long incubations using 100 ml plastic syringes with a Luer-lock connection to the chamber valve. Dissolved oxygen, pH, temperature, and salinity were measured using a handheld non-submergible multiparameter sensor (ProDSS equipped with ODO Optical Dissolved Oxygen and EXO pH Smart Sensors; YSI, Yellow Springs, OH, USA) attached to a kayak for analysis of samples within 1 minute of extraction. Chambers were flushed between incubations times by lifting them from the substrate and releasing the incubated water. A few minutes were allowed for any stirred-up sediments to re-settle before reassembling the chamber. Repeated measurements ($n = 15$) of chamber volume were conducted in shallow (<1 m) of Akumal Bay by filling chambers using a 100 ml syringe and recording variability between volumes to estimate error in volume measurements.

## Adaptations of the chamber

A more streamlined, low-profile, dome-shaped chamber (Fig. 1B) was used due to higher surge at the coral reef site, nearer to the reef crest than the protected seagrass area (Fig. 1D). This was also an adaptation for the lower profile of reef patches (~15 cm height) compared to seagrasses (~30 cm height). The panels were cut as curved triangles (29 × 16 cm, H × W) with arching sides and each dome was constructed of 8 panels and two 8 cm skirts were added at the base. These skirts were added due to higher wave action at this site, to prevent leakage around the seal. All plastic seams were sealed with a double seal 1 cm apart on each joint, using a generic, locally available heat sealer. For the high-energy environment (case study 2) a heavier base was created by adding 8 kg of fishing weights into the bicycle tyre tube to total ~10 kg per base. Chambers were randomised over the substrate types for each incubation; algae-, coral-, and sand- dominated. Incubations lasted 3.5 h. PAR was measured as described above in both case studies.

## Case study 1: Seagrass production

Benthic chambers ($n = 5$) were assembled and deployed over seagrasses and sediment substrate at 1.5 to 2 m depth. Seagrass surveys were conducted using the methods outlined
in *Hernández & van Tussenbroek (2014)* to estimate seagrass abundance in Akumal Bay (Fig. 1D). Sampling was conducted at the start and end of incubations at midday, late afternoon, and after dark. Each incubation lasted 1.5 to 2.5 h. The chamber constructed for the seagrasses (Fig. 1C) was taller and narrower than the chamber used in case study 2 (Fig. 1B) to accommodate the height of the seagrasses.

## Case study 2: Production rates of distinct coral reef communities

Benthic chambers ($n = 9$) were deployed by SCUBA divers at 2–3 m depth over 4 days for multiple incubations during solar noon and after dark over small patches of reef. Using swim patterns from a central point ('Case study 2' label on Fig. 1D), small patches of mixed coral (*Porites spp.*), algae, crustose coralline algae, and sand/sediment were identified ($n = 21$), of which 9 were randomly selected for incubation. Surface area and volume of the incubated reef patches were measured using photogrammetry. 3D models of incubated reef patches were constructed from ~100 photographs of each patch using Agisoft Metashape and compared following the methods outlined in *Lange & Perry (2020)*. Surface areas ($m^2$) and volume ($m^3$) were extracted from the 3D models. We converted the volume to litres and deducted it from the chamber volume to calculate the individual sea water volume for each incubation and used this, with the surface area measurement to normalise metabolic rates. All areas of flat sediment were measured from top-down photos using ImageJ (*Schneider, Rasband & Eliceiri, 2012*).

## Metabolism calculations

Net community production (NCP, mmol $m^{-2}$ $h^{-1}$) was calculated from changes to dissolved oxygen (DO) using the following equation:

$$NCP = \frac{\Delta DO \times V}{SA \times T} \tag{1}$$

where $\Delta DO$ is the change in dissolved oxygen (mmol $L^{-1}$), which is normalised to the chamber volume (V) of water in litres, the surface area (SA) of the incubated sample in $m^2$, and the incubation duration time (T) in hours. Surface area was measured from 3D models using the program Agisoft following the protocol outlined in *Lange & Perry (2020)*. Individual chamber seawater volumes (V) were calculated by converting the estimated organism volume extracted from 3D models ($m^3$) into litres (L) and subtracting this from the empty chamber seawater volume. Dissolved oxygen fluxes were normalised to individual chamber water volumes and organism surface areas.

## Photosynthesis-Irradiance models

Rates of productivity (NCP) were modelled to light using the hyperbolic tangent equation of *Jassby & Platt (1976)*:

$$P_{net} = P_{max} \times \tanh\left(\frac{\alpha \times E}{P_{max}}\right) + R \tag{2}$$

where $P_{net}$ is the modelled net production rate (mmol $m^{-2}$ $h^{-1}$), R is the average dark respiration rate (mmol $m^{-2}$ $h^{-1}$), and E is irradiance in PAR (μmol photons $m^{-2}$ $s^{-1}$).

The coefficients derived from the model include: the initial slope ($\alpha$) between NCP and light and the maximum gross photosynthetic rate ($P_{max}$).

## Statistical analysis

Statistical analyses were carried out using RStudio version 1.4.1717 (*R Core Team, 2021*) with the package *Tidyverse* (*Wickham, 2019*). Each data set was assessed for parametric assumptions using Shapiro Wilkes, Q–Q plots, and histograms. Data did not always meet the assumptions of normality; therefore, nonparametric statistical tests were selected for the analyses. For testing of chamber parameters (light, pH, temperature, leakage), paired sign tests and Wilcoxon rank sum tests with continuity correction were used, and for the case studies Kruskal Wallace tests were used to compare mean rates of NCP at different times of day and between different benthic communities. Linear regressions of seagrass NCP with light were modelled using *ggplot* with *ggpmisc* extension on R (*Wickham, 2016*; *Aphalo, 2021*). Photosynthesis - irradiance curves were modelled using R nonlinear least squares estimation and coefficients/model fit were evaluated based on $R^2$, confidence intervals, and standard error of the regression (sigma, $\sigma$).

## RESULTS

### Construction of benthic chambers for field deployment

Benthic chambers ($n = 6$) were successfully constructed and deployed by citizen scientists in the seagrass meadow of Akumal Bay in July 2019. Materials were sourced locally and the cost per chamber was USD $17.62. Details on costs for the items are included in Table 2. The time taken to fill rubber tubes with sand and weights, cut the plastic bag, and install a valve ranged from 15 to 30 min for 2 people and 45 min for 1 person. The chambers were deployed without difficulty by a range of users of varying experience.

### Leakage

Salinity inside the chamber at the start of the incubations ($38.9 \pm 0.3$ ppt; mean $\pm$ SD) was reduced by $0.8 \pm 0.3$ ppt over the course of the 4-h incubations. Overall, the difference in salinity was $-2.1 \pm 0.8\%$ between the start and end values of 3 replicate incubations during 4-h incubations. The salinity of the water outside the chamber remained stable throughout the incubations ($26.1 \pm 0.1$ ppt).

### Transparency

Laboratory testing found that light transmission through the polyethylene plastic was 74.4% at 750 nm and 61.1% at 400 nm (Fig. 2A). In the field, the difference between PAR measured simultaneously inside and outside the chamber over 3 field deployments was $13.0\% \pm 1.9$ (mean $\pm$ SD) and the difference was only found to be significant over 2 h at peak sun on 2 days (Wilcoxon signed rank test with continuity correction, Table S1).

### Temperature

Seawater temperatures measured inside ($30.7 \pm 0.7$ °C, median $\pm$ SD) and outside chambers ($30.7 \pm 0.8$ °C, $n = 9$) at the end of 3-h incubations in peak sun were not significantly different (Sign test statistic = 18, df = 28, $p = 0.185$).

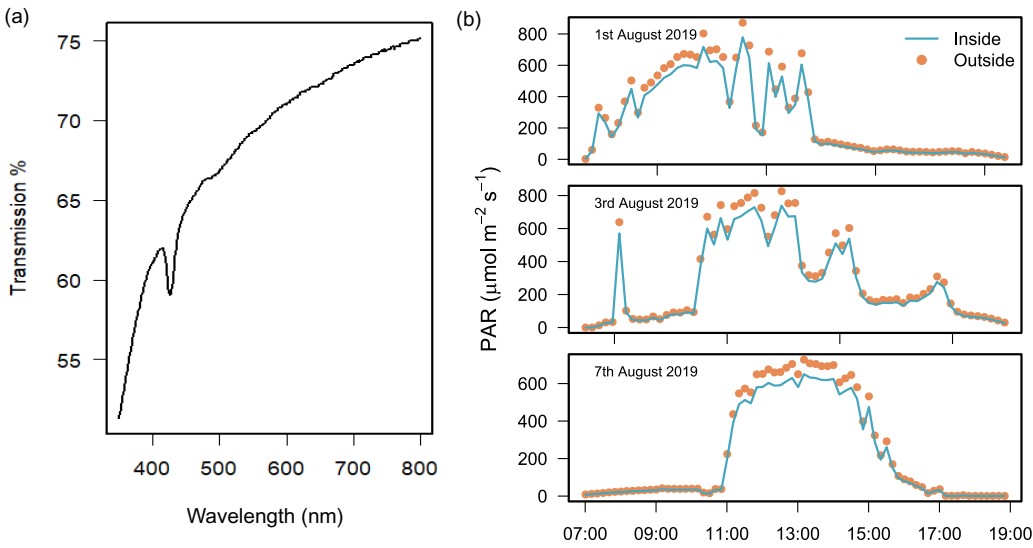

**Figure 2 Light transmission of the benthic chamber.** Chamber transparency measured with (A) a spectrophotometer to quantify percentage (%) transmission of light between 400 and 750 nm, and (B) light sensors during the field deployment of chambers ($n = 3$) to measure PAR ($\mu$mol m$^2$ s$^{-1}$) inside (blue line) and outside (orange dots) of the chamber over 3 days of exposure to natural solar radiation between 07:00 and 19:00 h.

## Case Study 1: Seagrass productivity rates increase with peak sunlight

Six benthic chambers were deployed over seagrass-sediment substrates in Akumal Bay. Seagrass coverage was composed of three species: *Thalassia testudinum*, *Syringodium filiforme*, and *Halodule wrightii*, and visual surveys indicated 50% areal coverage within the incubation area. Increases in DO concentration demonstrated positive net community productivity (NCP) during daylight incubations, while negative NCP after sunset indicated net respiration for all seagrass incubations (Fig. 3). All values presented as mean ± SD. The difference in dissolved oxygen (ΔDO) was 78.5 ± 7.5 $\mu$mol L$^{-1}$ at solar noon when concentrations of 272.5 ± 7.5 $\mu$mol L$^{-1}$ DO were reached and during night incubations fell to 186.8 ± 12.7 $\mu$mol L$^{-1}$ DO with an average ΔDO of −45.2 ± 12.7 $\mu$mol L$^{-1}$. NCP was significantly different during different times of the day (KW = 13.3, df = 2, $p = 0.00126$). Average NCP was highest at solar noon (6.7 ± 1.3 mmol m$^{-2}$ h$^{-1}$), lower in the late afternoon (2.4 ± 0.6 mmol m$^{-2}$ h$^{-1}$), and negative after sunset (−3.3 ± 0.4 mmol m$^{-2}$ h$^{-1}$). Seagrass NCP had a strong linear relationship with PAR ($R^2 = 0.93$, $p = <0.0001$). One incubation done over sediments ($n = 1$) also had the highest NCP rate at solar noon (2.6 mmol m$^{-2}$ h$^{-1}$) and was negative at night (−0.6 mmol m$^{-2}$ h$^{-1}$).

## Case Study 2: Measuring productivity of coral-algae reef patches

Compositions of 9 small patches of reef (planar surface area = 1,075.2 cm$^2$) were described by relative proportions of functional groups such as coral, macroalgae, crustose coralline algae, and sediment measured from 3D models (Figs. 4 and 5). Two patches consisted of sediments only, 3 were algae-dominated, and 4 were coral-dominated. One of the coral-dominated patches consisted of a coral, which had begun to bleach. Surface areas

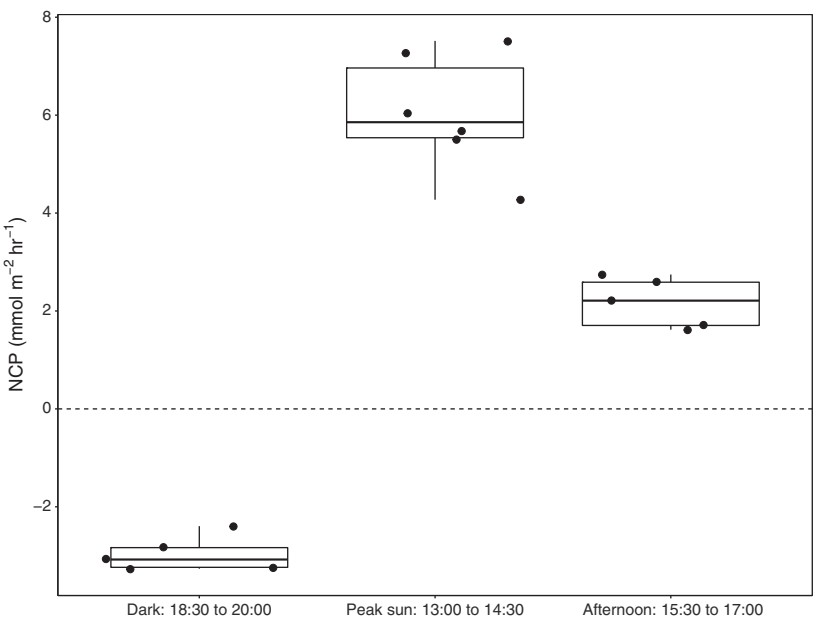

**Figure 3 Net community productivity (NCP) of incubated seagrass patches measured from benthic incubations.** NCP was calculated from the change in dissolved oxygen (ΔDO) normalised to chamber volume and incubation time (mmol m$^{-2}$ h$^{-1}$). Incubation chambers were deployed simultaneously over seagrasses in the dark ($n = 5$), at solar noon when PAR = mean $757 \pm$ SD 88 μmol m$^{-2}$ s$^{-1}$ ($n = 6$), and in the late afternoon, PAR = mean $463 \pm$ SD 75 μmol m$^{-2}$ s$^{-1}$ ($n = 5$).

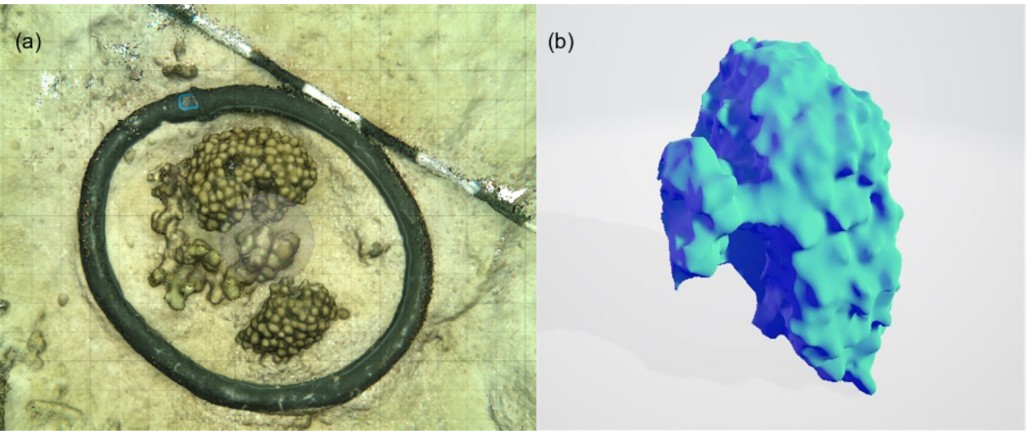

**Figure 4 An example of the 3D model generated for each of the incubated patches to measure surface area of complex structures and to estimate the volume occupied by the organism.** 3D models were built from photographs taken *in-situ* to estimate the composition, and measure volume and surface area of each patch: (A) screenshot of 3D model built from 85 photographs used for visual assessment of the relative percent cover of coral/algae *etc.* (B) a 3D profile of a coral colony from which volume and surface areas were derived. The surface area of the base of the colony was removed from the overall surface area calculated. Individual chamber seawater volumes were calculated by converting the estimated organism volume extracted from 3D models and subtracting this from the empty chamber seawater volume.
                                                           

calculated from 3D models were $2.86 \pm 1.04$ m$^2$. Chamber seawater volumes averaged (mean ± SD) $15.72 \pm 2.86$ L (Table 3). NCP was generally positive during daytime incubations and negative at night for all substate types (Kruskal–Wallace statistic = 49.9,

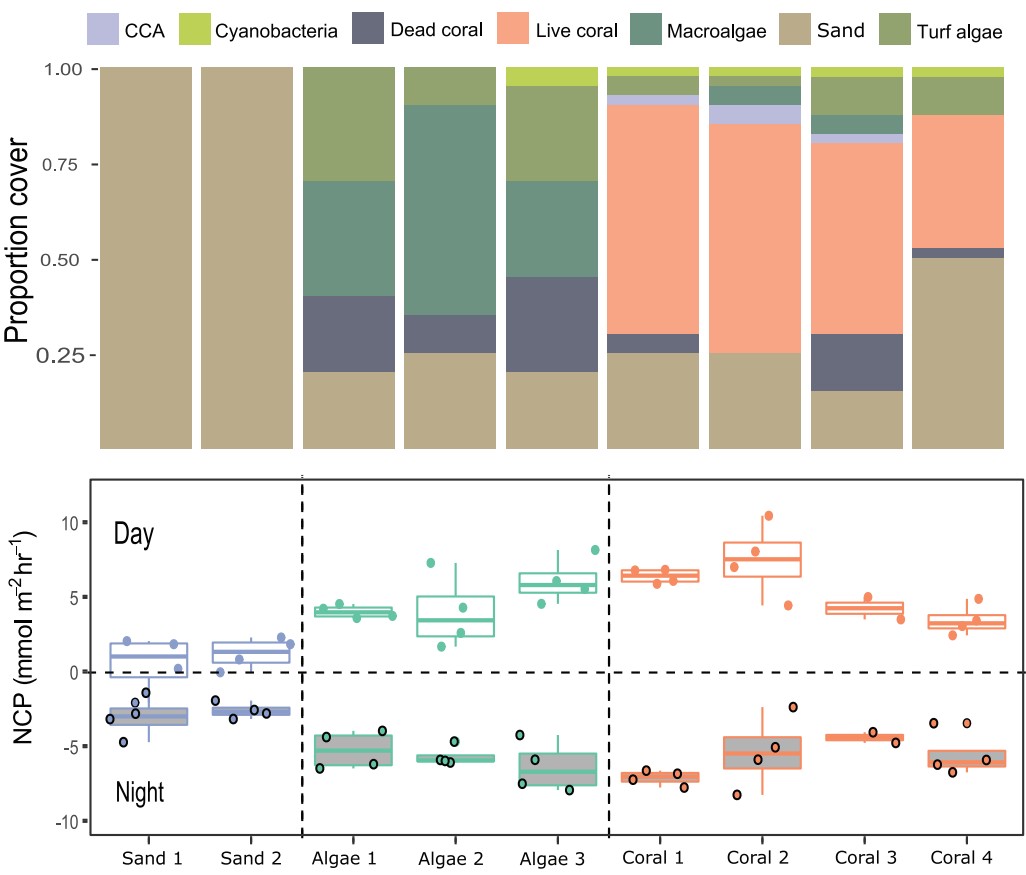

**Figure 5  Net community productivity (NCP) per incubated patch over 4 days and nights.** Stacked bar plots (top) show the proportional cover for each category of functional group and the boxplot (bottom) show the rates of NCP (mmol m$^{-2}$ h$^{-1}$) measured from differences in dissolved oxygen ($\Delta$DO) normalised to chamber volume and organism surface area (m$^2$) per hour following Eq. (1).

**Table 3  Measurements of benthic chamber and incubated coral reef patches.** Surface areas in square metres (m$^2$) and sea water volume in litres (L) of incubated reef patches estimated from 3D models following the protocol outlined by *Lange & Perry (2020)*. Substrate type refers to the pre-dominant functional group.

| Patch | Substrate type | Replicate | Surface area (m$^2$) | Seawater volume (L) |
|---|---|---|---|---|
| 1 | Sand | 1 | 1.14 | 20.00 |
| 2 | Sand | 2 | 1.22 | 20.00 |
| 3 | Algae | 1 | 2.24 | 13.26 |
| 4 | Algae | 2 | 2.43 | 15.55 |
| 5 | Algae | 3 | 3.20 | 12.82 |
| 6 | Coral | 1 | 1.77 | 15.41 |
| 7 | Coral | 2 | 2.73 | 16.20 |
| 8 | Coral | 3 | 1.03 | 17.27 |
| 9 | Bleached coral | 4 | 4.39 | 11.77 |

**Table 4 Comparison of net community production (NCP) rates between coral reef patches using non-parametric Wilcox pairwise comparisons.**

|  | Group 1 | Group 2 | $n_1$ | $n_2$ | Statistic | P value | Adjusted p value | Significance |
|---|---|---|---|---|---|---|---|---|
| Day | Algae | Coral | 12 | 14 | 67 | 0.4030 | 0.4030 | ns |
|  | Algae | Sand | 12 | 8 | 92 | 0.0002 | 0.0004 | *** |
|  | Coral | Sand | 14 | 8 | 112 | 0.0000 | 0.0000 | **** |
| Night | Algae | Coral | 12 | 14 | 91 | 0.7420 | 0.7420 | ns |
|  | Algae | Sand | 12 | 8 | 4 | 0.0002 | 0.0006 | *** |
|  | Coral | Sand | 14 | 8 | 8 | 0.0004 | 0.0008 | *** |

Note:
Rates of NCP (in mmol m$^{-2}$ h$^{-1}$) were compared between the different types of reef patches incubated in case study 2 (coral-, algae- and sand-dominated substrates) at day and at night. Significance level represented by stars (*) refers to the $p$-adjusted values, and 'ns' is an abbreviation for not significant.

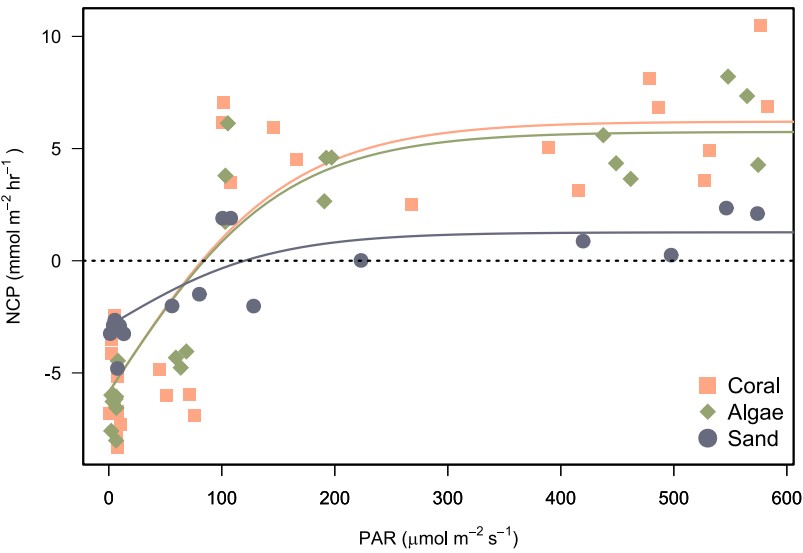

**Figure 6 Photosynthesis-irradiance curves fitted for net community production (NCP) of incubated communities of coral-, algae-, and sand-dominated reef patches.** NCP (mmol m$^{-2}$ h$^{-1}$) in relation to PAR (μmol photons m$^{-2}$ s$^{-1}$). Shapes and colours represent the different types of patch incubated. Models were fit by non-linear least squares using the hyperbolic tangent function (Eq. 2).

df = 1, $p = 1.61 \times 10^{-12}$). NCP rates measured from the 9 distinct patches were different during the daytime (Kruskal–Wallace statistic = 25.0, df = 8, $p = 0.002$) and at night (Kruskal–Wallace statistic = 19.2, df = 8, $p = 0.014$). When the patches were categorised into algae-, coral-, and sand-dominated groups, pairwise comparisons revealed that NCP was different between sand–algae and coral–algae, but algae and coral did not have significantly different NCP during the day or at night (Table 4).

Photosynthesis-irradiance data fit a hyperbolic relationship with PAR for all substrates (Fig. 6). Photosynthetic maximum ($P_{max}$) was highest in the algae-dominated substrate type (11.6 ± 0.9 mmol m$^{-2}$ h$^{-1}$) and lowest for sand/sediments (4.2 ± 0.6 mmol m$^{-2}$ h$^{-1}$). All model coefficients were significant (Table 5).
**Table 5 Summary of photosynthesis-irradiance curves modelled from coral reef net community production data using the hyperbolic tangent function (*Jassby & Platt, 1976*).**

|  | $P_{max}$ | ±SE | CI 2.5% | CI 97.5% | $p$ | $\alpha$ | ±SE | CI 2.5% | CI 97.5% | $p$ | $E_K$ | RSS | $\sigma$ |
|---|---|---|---|---|---|---|---|---|---|---|---|---|---|
| Algae | 11.60 | 0.86 | 9.82 | 13.38 | $3.96 \times 10^{-12}$ | 0.08 | 0.01 | 0.05 | 0.10 | $4.08 \times 10^{-06}$ | 151.83 | 96.98 | 2.10 |
| Coral | 12.08 | 1.13 | 9.77 | 14.40 | $4.80 \times 10^{-11}$ | 0.08 | 0.02 | 0.04 | 0.11 | $1.41 \times 10^{-04}$ | 155.07 | 258.25 | 3.15 |
| Sand | 4.17 | 0.64 | 2.79 | 5.55 | $1.42 \times 10^{-05}$ | 0.03 | 0.01 | 0.01 | 0.05 | $9.62 \times 10^{-03}$ | 139.65 | 24.18 | 1.31 |
| All | 10.13 | 0.66 | 8.81 | 11.45 | $1.91 \times 10^{-23}$ | 0.07 | 0.01 | 0.05 | 0.08 | $4.19 \times 10^{-09}$ | 154.62 | 510.58 | 2.78 |

**Note:**
Substrate-specific coefficients for the photosynthesis-irradiance models are displayed with standard error (±SE) and 2.5% and 97.5% confidence intervals (CI). The light saturation point (EK) was calculated by dividing maximum gross photosynthesis ($P_{max}$) by alpha ($\alpha$), the initial slope between NCP and light. Residual Sum of Squares (RSS) and standard area of the regression ($\sigma$) calculated to evaluate model fit.

## DISCUSSION

We designed and tested a novel benthic incubation chamber, drawing together key principles from existing designs (*e.g.*, *Yates & Halley, 2003*; *Camp et al., 2015*; *Roth et al., 2019*). The benthic chamber was designed to incubate small communities, single organisms, and sediments. The equipment created was low cost, reproducible, minimally invasive, and adaptable, with comparable features and capabilities to other *in-situ* benthic incubation chambers (*see* Table 1). The chambers were successfully assembled and deployed for fieldwork by citizen scientists and students. Construction of a single benthic chamber took between 15 and 30 min for two people, including the time taken to measure the volume of the chamber and to fill the tube base with sand. In the water, deployment over complex substrate took 5 min or less by two qualified divers.

The performance of the chambers was evaluated by conducting productivity measurements using field deployments of the chambers over over a seagrass bed (case study 1), as well as algae, coral, and sand dominated reef patches (case study 2). The results obtained using this method were comparable to those obtained in previous studies using established chamber designs, demonstrating the effectiveness and feasibility of the chamber design as a low-cost alternative for benthic incubations. Lowering the costs of *in-situ* incubation apparatus broadens the accessibility and scope for measurements of benthic metabolism for conservation monitoring purposes.

### Evaluating the chamber design

The benthic chamber design presented in this study is novel in its accessibility and scope for implementation with limited resources. Benthic incubations allow for direct measurements of benthic metabolism and are an important tool for quantifying carbon fluxes from physiological processes within benthic communities. However, the most inexpensive chambers currently available for *in-situ* community incubations cost around USD $200 to build and require bespoke parts and a submersible pump for operation (*Roth et al., 2019*), while the economical 'flexi-chamber' (*Camp et al., 2015*) is limited to individual coral colonies, rather than benthic communities. The chamber presented in the current study incubates communities or individuals and it costs <USD $20 to construct. At such low costs, multiple replicates are feasible. The training required to build, deploy, and collect data using most chamber designs is not yet widely available.

We demonstrated that the flux measurements can be normalised to surface area and volume data either using more traditional estimates of planar area and percentage cover in case study 1 or using a 3D modelling approach as in case study 2. This demonstrates the adaptability of our chamber design for example when the software and computational power required for the latter technique may not be available. The benthic chamber itself does not require any specialised equipment or training to deploy as materials were locally sourced and citizen scientists were able to make and deploy the chambers with minimal training.

We demonstrated that the novel chambers can be constructed from locally sourced materials without the need for any specialised parts. Plastic bags, bicycle inner tubes, fishing weights, and SCUBA gear for underwater transport were all easily sourced on site. We purchased Luer-lock valves from a local medical supply store, however, we also trialled the chamber construction and deployment with a plastic sports bottle cap instead of the valve (Fig. 1C). The chamber in this study was constructed for <USD $20, however it should be noted that the sensors used to measure dissolved oxygen, pH, and salinity inside the chambers should be considered as an additional cost. This is true of all but one of the selected chambers in Table 1 and would be dependent on the individual experiment being conducted with the benthic chamber and the precision and accuracy required.

To verify the reproducibility of the chamber design for determining photosynthesis, we conducted net community productivity measurements. First, we measured light attenuation of the polyethylene tent and found a reduction of 13% ± 1.9 between inside and outside the chamber (Fig. 2B). These results aligned with the transparency of comparable chambers, such as the 9% reduction in light reported in Roth et al. (2019) and 16% in Camp et al. (2015). Our results support previous findings that transparent plastic polyethylene or vinyl materials sourced at food packaging, hardware, or medical supply stores are effective for incubations of photosynthetic organisms (Yates & Halley, 2003; Camp et al., 2015). As the chamber is adaptable for distinct locations, uses, and resources, light attenuation should be measured given the variability in light transmission by different plastics.

One of the unique features of the novel chamber design is the malleable base, which facilitates deployment over hard, uneven substrate without using destructive methods to fix the chamber in place. This is particularly useful for incubations in managed or protected areas, where attachment by digging the chamber base into sediments or pinning the chamber to a coral reef using nails or cable tie attachments might not be feasible. The rubber tubing base was weighted with easily available materials to create a seal with the substrate, which could be adapted to 3D structure on the reef incurring minimal damage. Given that the malleable base could be a potential source of leakage, laboratory testing was conducted, and we demonstrated that the rate of water exchange was 2% over 4 h. We consider the novel benthic chamber to be a reproducible method and effective for accurate measurement, based on the results of laboratory testing and case studies. Water exchange during field deployments may vary, depending on the local conditions, for example surge, currents, and substrate type, therefore it is advisable that users of the

chamber incorporate a testing phase to adjust the weight of the base to ensure that it is sufficient to create a seal. Users could replicate the salinity method by adding hypersaline solution to the chamber and measuring salinity at the start and end of the incubation time (*Webb et al., 2021*), or an alternative if refractometers are not available would be the visual assessment of leakage using non-toxic food dye. It is also possible to visually assess water mixing and movement using dye. We injected red food dye into field-deployed chambers and observed full dispersal within one minute (Fig. 1C).

Water movement over the reef is a driver of metabolic fluxes (*Comeau et al., 2014, 2019*) and maintaining natural water movement without the restriction of a solid chamber wall sustains ambient *in-situ* conditions throughout the sampling period. The flexibility of the plastic bag material used for the tent enclosure also has the benefit of reducing costs compared to rigid chambers by encouraging natural water movement and mitigating the need for a submersible pump inside the chamber. The concept of using a flexible plastic enclosure was demonstrated with the Flexi-chamber design (*Camp et al., 2015*). While the flexibility of the tent enclosure promotes natural water movement and mixing, any benthic chamber will disrupt natural water movement, which should be considered when interpreting data collected with benthic chambers in general. The NCP rates measured in our case studies also demonstrate that water movement is maintained by the flexible walls, as supported by the NCP rates measured (discussed below).

The benthic chamber used in this study is adaptable to different substrate types, environmental conditions, and research applications. The simple and flexible design enables easy adjustment of size and volume, and additional valves or sampling ports can be easily installed. The chamber can be used to measure additional parameters to those measured in this study, for example, to collect sea water samples for carbonate chemistry to measure calcification rates. This may add to the expertise and costs of using the chamber. The weight of the base can be altered for high or low energy systems, and the height of the dome can also be changed to accommodate different sample sizes and heights. It can also be adjusted to enhance or minimise hydrodynamic mixing by naturally occurring currents or wave action. As the base is malleable, it can be used on hard substrate or soft sediments as demonstrated in our case studies over reef communities and seagrass beds. The two case studies demonstrated the adaptability of the chamber.

## Seagrass productivity measured with the benthic chambers

Incubations over seagrass and sand demonstrated a typical diel trend of NCP increasing with sunlight and switching to net respiration at night. Seagrass productivity rates were in line with those of previous studies measuring seagrass NCP, however, variability does exist (*Duarte et al., 2010*). The average solar noon NCP rate measured in this study ($6.7 \pm 1.3$ mmol m$^{-2}$ h$^{-1}$) was lower than NCP measured in Florida with the SHARQ chamber ($12.3 \pm 1.0$ mmol m$^{-2}$ h$^{-1}$, *Turk et al., 2015*). However, our rates were higher than the average NCP for tropical Western Atlantic seagrass meadows described in a meta-analysis by *Duarte et al. (2010)* in which daily NCP averaged $23.7 \pm 7.8$ mmol m$^{-2}$ day$^{-1}$ over 155 seagrass studies, which would equate to an hourly rate of approximately 2.2 mmol m$^{-2}$ h$^{-1}$. Seagrasses in Akumal Bay are colonised by epiphytes and cyanobacteria

(*Hernández & van Tussenbroek, 2014*), associated with high nutrient load in the bay, which may have influenced our NCP measurements (*Coleman & Burkholder, 1994*; *Borowitzka et al., 2007*). Productivity increased linearly with light and did not reach photosynthetic saturation. Therefore, it was not possible to fit the hyperbolic tangent equation, possibly due to the limited light conditions in the study design (*i.e.*, there were only two light levels).

### Measuring productivity of coral–algae reef patches

Net production rates measured for all reef patches were positive during light incubations and negative for dark incubations in line with previous studies of *in-situ* coral reef metabolism measured with benthic chambers. For example, light NCP measured with the SHARQ enclosure at a reef site in Florida averaged $8.6 \pm 1.0$ mmol m$^{-2}$ h$^{-1}$ (*Turk et al., 2015*), a rate that also aligned with NCP measured estimated by CROSS (*McGillis et al., 2011*), and previous SHARQ deployments within the Caribbean region (*Turk et al., 2015*). Our night respiration rates, measured from around dusk until 21:00–22:00 h, were also in agreement with the values measured in the early evening using SHARQ deployments in *Turk et al. (2015)*. The highest rates of NCP were measured in coral patch 3, which was the only replicate containing *P. astreoides* as well as *P. porites*. Net community production was lowest in patch containing a bleached *P. porites* colony ('Coral 4' on Fig. 5), which had been subjected to heavy sedimentation and algal overgrowth. The lower rates of NCP measured in the incubations of this colony most likely reflect lower metabolic rates due to environmental stress. Coral patches 1 and 2 were very similar in terms of the health, size, and structure of the *P. porites* colonies incubated and they had minimal (<10%) algae within the chambers, as reflected by the similar rates measured in these incubations. We modelled reef NCP data to light and found a hyperbolic relationship between PAR and NCP as have other studies (*Long et al., 2013*; *Takeshita et al., 2016*; *Turk et al., 2015*). However, our modelling approach was somewhat limited by the range of light levels and number of incubations. Future research should aim to include more incubations at different times of day. The low costs of the chamber we present in this study supports such research.

## CONCLUSIONS

Monitoring of coastal ecosystem health is of critical importance for tracking and predicting ecological response to global climate change and other localised threats. This study presents a novel benthic chamber design, which reduces the cost of *in-situ* incubations of benthic organisms in shallow coastal ecosystems, while maintaining reproducibility. The chamber was designed to be minimally invasive and adaptable, which we demonstrate through successful field deployment. We measured productivity rates for seagrasses and reef patches in line with previous studies and provide a comparison of rates. Further studies are needed for quantification of coastal carbon cycling and efficient methods to enhance conservation monitoring, and the low-cost benthic chamber we describe overcomes some of the limitations of other designs. It is a potential tool for diverse users to employ in such research endeavours.

# ACKNOWLEDGEMENTS

We thank Dr. Karl Attard, Dr. Alireza Merikhi and one anonymous reviewer for their constructive feedback to improve upon the original manuscript. Data collection, logistics and planning were conducted with the support of Operation Wallacea scientists Sabrina Weber, Jessica Rose-Innes, Holly Jones, Dr. Gemma Fenwick, Dr. Kathy Slater, and other staff and student research assistants who were generous with their time building and trialling the chambers. We thank the Akumal Dive Center's skilled team of divers, captains, and staff for their valuable practical advice and logistical support. Thank you to team Coralium, UNAM for help with sourcing local materials, to Charlotte Slaymark, Kenny Roberts, and Jessica Walker at the University of Glasgow for technical support in the laboratory and to the team at SCENE for facilitating field testing in Scotland. A special thank-you to the Hotel Akumal Caribe for hosting and to Leona Kustra and the Coral Conservation Society for facilitating this research.

## Funding

Fieldwork in Mexico was funded by Operation Wallacea. This project was supported by the Royal Geographical Society (with IBG) with a Henrietta Hutton Research Grant. Funding was also provided by the Coral Conservation Society. The funders had no role in study design, data collection and analysis, decision to publish, or preparation of the manuscript.

## Grant Disclosures

The following grant information was disclosed by the authors:
Operation Wallacea.
Royal Geographical Society.
Coral Conservation Society.

## Competing Interests

Anastazia Banaszak is an Academic Editor for PeerJ.

Dan A. Exton is employed by Operation Wallacea, and Lauren H. Donachie is a summer research assistant for Operation Wallacea.

## Author Contributions

- Jennifer Mallon conceived and designed the experiments, performed the experiments, analyzed the data, prepared figures and/or tables, authored or reviewed drafts of the paper, and approved the final draft.
- Anastazia T. Banaszak conceived and designed the experiments, authored or reviewed drafts of the paper, and approved the final draft.
- Lauren Donachie conceived and designed the experiments, performed the experiments, authored or reviewed drafts of the paper, and approved the final draft.

- Dan Exton conceived and designed the experiments, authored or reviewed drafts of the paper, and approved the final draft.
- Tyler Cyronak conceived and designed the experiments, authored or reviewed drafts of the paper, and approved the final draft.
- Thorsten Balke conceived and designed the experiments, authored or reviewed drafts of the paper, and approved the final draft.
- Adrian M. Bass conceived and designed the experiments, authored or reviewed drafts of the paper, and approved the final draft.

## Field Study Permissions

The following information was supplied relating to field study approvals (*i.e.*, approving body and any reference numbers):

Field experiments were approved by the National Commission of Protected Natural Areas of Mexico (Comisión Nacional de Áreas Naturales Protegidas, CONANP), Permit reference: F009.DRBCM/240/2019 dated 02 May 2019.

## Data Availability

The raw data is available in the Supplemental Files.

## Supplemental Information

Supplemental information for this article can be found online at http://dx.doi.org/10.7717/peerj.13116#supplemental-information.

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
