# Peer review of "A low-cost benthic incubation chamber for in-situ community metabolism measurements"

_PeerJ, doi:10.7717/peerj.13116_

## Round 0.1 · original submission · Minor Revisions

The results from the three reviewers have been received. All the reviewers gave your paper a very positive evaluation with numerous constructive and very useful comments. I myself agree with the three reviewers. Congratulations! My comments has also shown below: Line 27-29, please explain why you judged the leakage (2.08 ± 0.78 % water exchange over 4 hours) and light transmission (86.9 ± 1.9 %) acceptable.

·

Basic reporting

Language is clear and unambiguous. Professional English is used throughout. Literature references are adequate and up-to-date. Figures and tables are informative and clear. Raw data shared. Article is self-contained and results are relevant to research question.

Experimental design

The research presented is original and within the Aims and Scope of PeerJ. The research question is well defined, relevant and meaningful. It is stated how the research fills an identified gap in methodology. The research is of a high technical and ethical standard. Investigation could have been performed more rigorously, by performing additional analyses. Methods are described with sufficient detail to replicate.

Validity of the findings

All underlying data have been provided. Annotation of headers could be improved to be made more descriptive, and include units. Conclusions are well stated, linked to original research question and mostly supported by results.

Additional comments

The paper by Mallon and coauthors presents a benthic incubation chamber for measurement of community metabolism. While many chamber designs exist, this chamber design is different because it is based on materials that are low-cost and are easy to source. The authors test the design in the laboratory and in the field through performing incubations on sediments, seagrasses, and coral reefs.
Overall, I appreciate the effort to make such a technology more accessible. The paper itself is well-written, and figures are for the large part clear and relevant. The flux dataset seems to be of a high quality. Please find my main comments to the paper as well as detailed comments below.
Main comments
1. Description of chamber performance: The authors do a good job describing light attenuation inside the chamber bags. There are two points that could be improved:
a. Description of mixing inside the chamber: I appreciate that bags allow some water movement inside the chamber and this is superior to having a Plexiglas chamber with no stirring. However, I suspect that gradients will build up inside the bags, so the resolved flux will be dependent upon the location of the valve. This is my main concern with the design of the chamber, and it is unfortunate that the authors do not quantify this directly. Perhaps the authors can expand on this point, based on previously published work using similar bags? Similarly, one concern I have with applying chambers in highly productive habitats is the build-up of supersaturated or hypoxic conditions during light and dark incubations. Since no data are presented on this, I am curious to know the highest and lowest O2 values observed inside the chamber, and how this compares to what can be expected for these habitats to experience naturally?
b. Leakage tests were not performed under field conditions. There are straightforward ways to assess leakage, which the authors describe, but this was not done. I do not think that the laboratory conditions tested are representative of many of the habitats studied. If the authors could perform a field test, then this will improve design validation.
2. The premise of the paper seems to be low-cost and accessible technologies. I agree that this is the case with the chamber itself. However, the 3D modelling used to reconstruct benthic surfaces to calculate reef surface area and volume is far from trivial, and requires specialized expertise, software, and high-performance computers. Since these measurements are used to normalize metabolic rates, it seems that it is hard to argue that this is overall a low-cost and accessible method. I must say that I fully appreciate the advantages of photogrammetry and what this brings to metabolic studies, but I hope the authors are able to see these conflicting points. In principle, you could normalize to reef planar surface area and still obtain a flux, although this will of course not be as informative. I believe the text should be adjusted to better represent this.

Detailed comments
L23: I do not think that chambers are non-invasive. There are true non-invasive methods out there such as eddy covariance and gradient methods.
L38: Sequestration. Replace with ‘cycling’. Sequestration refers to organic matter buried deep in sediments, which is not the focus of the study.
L46: Again here, focus on cycling rather than sequestration.
L48: One can calculate more than the NCP from O2 flux measurements: photosynthesis (gross and net), respiration, and the balance between the two (NCP).
L51: ‘autonomous sensing’. I suggest replacing this with ‘benthic boundary layer approaches’
L52: Berg et al. 2019. A better reference is Berg et al. (2022) Aquatic Eddy Covariance: The Method and Its Contributions to Defining Oxygen and Carbon Fluxes in Marine Environments. Annual Review of Marine Science.
L52: ‘Flow respirometry’. I suggest replacing this with ‘Eulerian approaches’.
L56: I don't think that chambers can be easily deployed on rocky surfaces such as corals. I suggest rewording to say 'have been deployed'.
L105: Again here, I question whether chambers are non-invasive.
L138: Replace ‘course’ with ‘coarse’
L138: Was there any flow inside the aquarium, as you’d expect to find in situ?
L175: Was this done in the field or lab?
L185: I think ‘join’ should be ‘joint’
L198: How was this deduced?
L201: Did individual deployments last four days?
L203: What is CCA?
L208: How did you convert from volume or surface area to biomass?
L218-219: Wasn’t this done using the 100 ml syringes, as described in L174-176?
L251: You should specify that this was done in the lab.
L260-261: It really depends how significance was tested. I find it hard to believe that there was no significant difference during the high light periods. The take home message is that there is some light attenuation, not that there isn't any.
L263: It is not clear to me why you performed this test. Is it because you thought that water inside the chamber might heat up relative to the outside? What are the reasons/mechanisms for this? It would be useful to include a sentence in the intro or methods about why this may be important.
L270: It would be useful to know how much O2 increased/decreased during the incubation. One of the drawbacks of chambers is that O2 can become strongly supersaturated or turn hypoxic, which would influence metabolic processes.
L284-287: This should be in the methods section.
L303: Here too. Chambers are not non-invasive.
L313-314: The chambers are low-cost, but the overall approach requires specialist expertise, software, and computational power.
L328: I agree that the chambers can be deployed by non-experts, but the data processing presented herein, particularly related to 3D models of benthic communities is far from trivial. I think this needs to be represented a bit better.
L352: Placing a chamber onto organisms is always going to be invasive. For instance, the rubber tubing base on the seafloor will curb the supply of oxygen to the sediments, which if left for long enough will turn the surface sediments beneath the tube anoxic. I can imagine that the base will move around some in a swell or current too, causing abrasion. So the chambers likely have some impact, albeit small.
L357: ’without causing damage’. Again here, perhaps say ‘incurring minimal damage’
L360: effective for accurate measurement. It should be stated that this was done under laboratory conditions.
L369: I highly doubt the bags maintain natural water movement. How do you know this is the case?
L372-373: Again here, you did not test it, so you do not know this is the case.
L373: I do not see how this was demonstrated in the paper by Camp et al. 2015. Can you expand?
L376-384: This paragraph is mostly a repetition. Can be greatly reduced or removed.
L388: Did you measure seagrass shoot densities? The Duarte paper is a global compilation, it would be more interesting to compare like with like.
L391-394: If I understood this correctly, you are comparing NCP at solar noon with average daily NCP. It is not the same thing.
L408: In what way/s are they in agreement?
L412-413: More eutrophic sites might have higher rates of O2 consumption at nighttime. Was this the case?
Figure 2: Panel labels missing
Figure 3 and Figure 4: to my understanding, they contain the same data, so I suggest keeping one of them (I personally prefer Fig. 4).
Figure 7: x-Axis label should be µmol m-2 s-1

·

Basic reporting

General comments:

This paper by Mallon et al. provides a new design for controlled studies of the benthic communities. In general, the proposed method in this paper can be used by many scientists and citizen-scientists to record flux values in locations with poor access to advanced methods and materials. Thus, I believe this proposed design can be very useful in special circumstances.
Light, waves, flow, and temperature are 4 factors that can significantly impact the benthic fluxes. Benthic enclosure methods in general interfere with these factors and thus these methods are usually called invasive or intrusive methods. The proposed method in this paper is therefore not excluded from these limitations. Considering the purpose of the paper that is to provide an accessible method for circumstances with limited resources, these limitations can be acceptable.
Flow can be recreated in a category of benthic chambers using stirring mechanisms. Including flow is fundamentally crucial for measuring benthic fluxes over many benthic communities such as permeable sediments. Mallon et al. paper does not provide information about these types of chambers (please see: Benthic metabolism and degradation of natural particulate organic matter in carbonate and silicate reef sands of the northern Red Sea by Huettel et al.). I think it is crucial for the reader to be aware of current state of technology before proposing a simplified version.
The language of the manuscript is clear, and the text is relatively cohesive with a good flow. There are a few suggestions in this regard that will follow.
In conclusion, this paper is a useful paper that tries to provide the scientific community with a handy solution where other more precise technologies are not accessible. I will include a few recommendations that can improve the accuracy of statements and the quality of the paper.

Specific comments to the authors:

table 1 please include benthic chamber design that I mentioned.
table 4 please clarify what the significance is describing. Various statistical test has different meanings.
table 5 you may transform this table to a very clear plot. Readers are mostly visual, and plots are better in general.
figure 2 the unit for light is umol photons m-2 s-1 please fix it here and on the plots. Why do you report transmission? please report the date of measurements on each graph of the panel b or in the caption.
figure 3 the unit in the caption does not match the unit on the graph. Usually, researchers report day and night NCP. I understand that you wanted to make 1.5h measurement times, but please change this plot to daytime and nighttime NCP.
figure 4 As you correctly mention in your other plot of Fig. 7 PAR-NCP should follow a nonlinear relationship. The reason for this linear relationship might be insufficient data-points. I recommend you to explain this in the text. Usually in these plots only the daytime data is reported, please remove the night data from the plot.
figure 7 This hyperbolic tangent function is called Michaelis–Menten function that explains productivity change with PAR. Please rephrase the caption and add the Michaelis–Menten regression equations on this plot. It is well possible to have negative fluxes during the daytime, but I think you included night data in this plot which must be excluded.

Line 22: Please change specialist equipment to specialized equipment here, and in other places.
Line 23: Please change non-invasive to another term as it usually refers to not interfering with natural conditions.
Line 51: autonomous sensing is not a common phrase used in the field. These are true noninvasive methods that does not interfere with light flow or hydrodynamic that are either boundary layer methods or eddy covariance methods. These measure metabolism of a large surface area. Please distinguish these in the text and cite a few papers for each of them
Line 52: I think flow respirometry is not usually used in the field for benthic flux measurements.
Line 53: please provide a few examples and citations for enclosed incubation methods.
Line 59: standard benthic chamber might have different designs for different purposes. You should look at the paper that I mentioned in my general comments to see what benthic chambers are referred to when studying metabolism of sediments.

Experimental design

Line 111: please explain the benthic ecosystem, fauna, and flora in your deployment site. Please report the deployment dates.
Line 122: Your experimental design does not accommodate mixing which is understandable with your goal of building an inexpensive and handy incubation method. Please explain this here or in the discussion.
Line 212: the metric unit for distance is meter please change the cm to m.
Line 215: In many benthic situations pumping action by waves and tides may change the volume of these flexible chambers. Please explain if you considered this in your calculations.
Line 227: please change the unit of PAR to the correct one.

Validity of the findings

This is very appropriate and useful to report two different cases for testing this method. Unfortunately authors did not compare the results of their methods with other available methods in the field, which is understandable because methods such as traditional benthic chambers or eddy covariance method might not have been accessible for the authors.

Line 368: Any enclosure flexible or rigid obstructs the flow. The flexibility might help but because pumping action and circulation is not guaranteed there is always a need for water circulation in the chamber. In permeable sediments flow impacts the fluxes that I doubt your design accommodates this efficient circulation. For this reason, your design is not considered a noninvasive design. You did not support your claim that the flexibility of the enclosure provides efficient circulation with sufficient evidence.
Line 399: there is a possibility that you did not have enough data point to fit on the Michaelis–Menten equation. I think you should rephrase the hyperbolic tangent equations to the Michaelis–Menten.
Line 419: I think your design is very practical but not very novel.
Line 423 and 425: As I understood you are trying to simplify an expensive method to make it available for special circumstances. Of course, always there are needs for further studies, but the benthic chambers are rather an older method for benthic flux measurements and an established technology. I suggest you rephrase the intention of your research here.

Additional comments

Thank you to the authors for their effort in providing an inexpensive method that helps increase the availability of benthic flux measurement methods.

Reviewer 3 ·

Basic reporting

The article is written in professional style throughout.
Literature references are sufficient and context is provided.

Experimental design

Methods (i.e., chamber construction and application) are not sufficiently described to replicate experiments. Please provide additional information. See comments below for details.

Validity of the findings

Conclusions are often rather broad and limitations of the work is not well discussed. See comments below for details.

Additional comments

The study presents a low-cost approach to measure benthic community metabolism in situ. Although neither the method nor the topics the paper touches are exceedingly novel, I believe that the creative idea may benefit coastal ecosystem studies where resources are limited. The authors present sufficient evidence for the method's functionality and explain well how to replicate measurements. I am optimistic that the approach will benefit coastal ecosystem studies where resources are limited. I only have a few comments to improve clarity and encourage scientific discussion.

1) Chamber construction and deployment description are incomplete in many instances. Please refer to line-by-line comments for specific sections. Make sure that the reader can replicate your procedures.
2) I suggest adding two sections to the discussion. First, please discuss the limitations of your work and the use of chambers in general. They are not the perfect methods for all types of applications. Second, please discuss how your chamber can be used for various applications and measurements. Here, you can add even more value to your work. Oxygen fluxes are interesting and show very well the functionality of the chamber. However, other measurements (e.g., nutrients, carbonate chemistry) etc. may be performed too, increasing the chambers' application range. On the other hand, some measurements may not be suitable for this type of chamber material (e.g., DOC measurements that are sensitive to organic carbon contamination by various types of plastic). Guiding the reader through the application ranges will improve the strength of your study.


Line-by-line comments:

18 – 20: Benthic incubations are "invasive", enclosing organisms from the surrounding water. I would specify the environmental variables here: i.e., temperature and light. On top, I suggest highlighting the most critical benefit of in-situ incubations: Using natural communities with all associated organisms rather than simplified, reconstructed communities used in ex-situ laboratory measurements.

23: Non-invasive only in the sense that you keep the incubated community intact. Nevertheless, the organisms are enclosed from their surroundings, critically changing the exchange of solubles with the surrounding water.

30: Patches of reef comprised of coral, algae and sand? Or did you measure all three types of substrates?

33: Please finish the abstract with a section including the primary take-home message, the additional findings of importance and a perspective on how your work will benefit others.

51: I would refer to autonomous sensing as Aquatic Eddy Covariance.

54: The argumentation here on why you tried to improve incubation approaches is weak. What are the benefits over the other methods? (e.g., other variables than O2 fluxes can be measured by discrete sampling, something that is not possible with AEC). Benthic incubations also have many limitations over the other approaches. I suggest being open with this throughout the manuscript and having a specific section on the "limitations" of this approach in the discussion.

63: Inserted where? Unclear here.

101: I would outline here or elsewhere in the introduction what size your chambers are able to cover and why this is a "good" choice. What is the issue with smaller or larger chambers?

104: You may want to add what this chamber is suitable of measuring.

122: I suggest adding the materials to the listing, otherwise the materials in the following lines (125 – 126) are without context to where they belong to.

125: What was the shape and size of the plastic bags?

129: I miss a more detailed description on how the chambers are put together. These details will help the reader to reproduce your design.

133: You use many synonyms for your plastic bag "water-tight enclosure" (l 122), "transparent polyethylene" (l 125), "plastic membrane" (l 133), "polyethylene tent" (l 343)… I suggest introducing a definition once and using one term throughout the manuscript to avoid confusion.

135 – 137: This is a non-quantitative approach without any scientific meaning.

140: On what substrate was the chamber placed? If only on the bottom of the tank, how can you be sure the sealing works in other environments, too?

145: I like the approach using salinity differences. Great idea!

159: Incubations ran for 4 days? Please clarify.

174: How were the chambers flushed? Lifting the base from the sediment causes disturbance through stirred up sediment.

174 – 176: How was the water volume within the chamber assessed? This does not become clear from this section but is crucial for correct estimates of water chemistry changes.

207 – 208: How did you calculate biomass of corals and algae from surface area and volume? The surface area depends strongly on your resolution of the raster of the 3D model; i.e., higher resolution = larger surface area. Volume calculations are only marginally affected by this.

211: How were incubations "initiated"? There is evidence that oxygen fluxes exhibit a lag after closing incubation chambers, especially during dark incubations (i.e., oxygen is increasing for a few minutes before respiration dominates). Did you consider using oxygen sensors inside the chambers rather than making start-end measurements? What would be the dis/advantages?

217: How was the volume within the chamber assessed exactly?

217: Did you use the 3D surface area of the incubated community or the planar reef area of the incubated reef patch? The difference in the outcome is detrimental for interpreting your results.

250: How did you assure that no sediments were stirred-up?

256: Please provide calculations/results of the water exchange rather than only referring to the change in salinity.

259: You are referring to Fig 2a here. In Fig 2a, I assume the x-axis label is wrong and should be the wavelength? In 2b, the x-axis label is missing and should, likely, be local time?

287: Again, wouldn't it make more sense to normalize fluxes to the planar reef area enclosed by the chambers? All other types of measurements (e.g., larger chambers, eddy covariance etc) are based on the planar surface area.

350: What is the impact of the chosen plastic on UV radiation? Please discuss how this may influence rates of photosynthesis for very shallow waters, where UV may inhibit some photosynthetic processes.

367: Please also discuss how the seal works on rugged, uneven surfaces in this section. There were no tests performed with this regard, weren't there?
375: Very good and important advance to other midsized chambers!!

Figures and Tables:

Figure 1: Please separate the background picture from the other figures to improve clarity. The most important parts of this figure are pictures b) and c) and the schematic d).

Figure 5: Is not mentioned in the text?

Figure 6: Is mentioned in the text after referring to Figure 7. Please rearrange the order

Figure 7: Is referred to in the text before Figure 6.

Figure S1: Is missing?

Figure S2: Is mentioned in the text but I cannot find the file to it. It is also referred to in the text before Fig. S2.

Table 3: What is the surface area referring to? Isn't the enclosed planar surface area around 0.1 m2?

---

## Round 0.2 · accepted · Accept

Great paper! I just recommend the inclusion of "Individual chamber seawater volumes were calculated by converting the estimated organism volume extracted from 3D models and subtracting this from the empty chamber seawater volume" also in the caption of Fig. 4.